# Anticancer, Anticoagulant, Antioxidant and Antimicrobial Activities of *Thevetia peruviana* Latex with Molecular Docking of Antimicrobial and Anticancer Activities

**DOI:** 10.3390/molecules27103165

**Published:** 2022-05-16

**Authors:** Aisha M. H. Al-Rajhi, Reham Yahya, Tarek M. Abdelghany, Mohamed A. Fareid, Alawlaqi M. Mohamed, Basma H. Amin, Abdurrahman S. Masrahi

**Affiliations:** 1Department of Biology, College of Science, Princess Nourah Bint Abdulrahman University, P.O. Box 84428, Riyadh 11671, Saudi Arabia; amoalrajhi@pnu.edu.sa; 2Medical Microbiology, College of Science and Health Professions, King Saud Bin Abdulaziz University for Health Sciences, Riyadh 11426, Saudi Arabia; yahyar@ksau-hs.edu.sa; 3King Abduallah International Medical Research Center, Riyadh 11671, Saudi Arabia; 4Botany and Microbiology Department, Faculty of Science, Al-Azhar University, Cairo 11751, Egypt; mohamedfareid73@yahoo.com; 5Basic Science Department, Prep Year Deanship, University of Ha’il, Ha’il 2440, Saudi Arabia; 6Biology Department, Faculty of Science, Jazan University, Jazan 45142, Saudi Arabia; mohamedalawlaqi@hotmail.com (A.M.M.); amasrahi@jazanu.edu.sa (A.S.M.); 7The Regional Centre for Mycology and Biotechnology (RCMB), Al-Azhar University, Cairo 11751, Egypt; basmafarag.18@azhar.edu.eg

**Keywords:** latex, molecular docking, antimicrobial, anti-inflammatory, anticoagulant, anticancer

## Abstract

Natural origin molecules represent reliable and excellent sources to overcome some medicinal problems. The study of anticancer, anticoagulant, and antimicrobial activities of *Thevetia peruviana* latex were the aim of the current research. An investigation using high-performance liquid chromatography (HPLC) revealed that the major content of the flavonoids are rutin (11.45 µg/mL), quersestin (7.15 µg/mL), naringin (5.25 µg/mL), and hisperdin (6.07 µg/mL), while phenolic had chlorogenic (12.39 µg/mL), syringenic (7.45 µg/mL), and ferulic (5.07 µg/mL) acids in latex of *T. peruviana.* Via 1,1-diphenyl-2- picrylhydrazyl (DPPH) radical scavenging, the experiment demonstrated that latex had a potent antioxidant activity with the IC_50_ 43.9 µg/mL for scavenging DPPH. Hemolysis inhibition was 58.5% at 1000 µg/mL of latex compared with 91.0% at 200 µg/mL of indomethacin as positive control. Negligible anticoagulant properties of latex were reported where the recorded time was 11.9 s of prothrombin time (PT) and 29.2 s of the activated partial thromboplastin time (APTT) at 25 µg/mL, compared with the same concentration of heparin (PT 94.6 s and APPT 117.7 s). The anticancer potential of latex was recorded against PC-3 (97.11% toxicity) and MCF-7 (96.23% toxicity) at 1000 μg/mL with IC_50_ 48.26 μg/mL and 40.31 µg/mL, respectively. Disc diffusion assessment for antimicrobial activity recorded that the most sensitive tested microorganisms to latex were *Bacillus subtilis* followed by *Escherichia coli*, with an inhibition zone (IZ) of 31 mm with minimum inhibitory concentration (MIC) (10.2 μg/mL) and 30 mm (MIC, 12.51 μg/mL), respectively. Moreover, *Candida albicans* was sensitive (IZ, 28 mm) to latex, unlike black fungus (*Mucor circinelloides*). TEM examination exhibited ultrastructure changes in cell walls and cell membranes of *Staphylococcus aureus* and *Pseudomonas aeruginosa* treated with latex. Energy scores of the molecular docking of chlorogenic acid with *E. coli* DNA (7C7N), and Rutin with human prostate-specific antigen (3QUM) and breast cancer-associated protein (1JNX), result in excellent harmony with the experimental results. The outcome of research recommended that the latex is rich in constituents and considered a promising source that contributes to fighting cancer and pathogenic microorganisms.

## 1. Introduction

Historically, latex-creating plants have been regarded as having significant therapeutic origin because of their characteristic ingredients. According to the previous report [1], approximately 10% of plants secrete latex, belonging to over 40 families such as Apocynaceae, Euphorbiaceae, Moraceae, Astereacea, Cannabinaceae, Papavereceae, Asclepidiaceae, and Sapotaceae. Numerous documents mention that latexes contain multifunctional bioactive constituents [2,3] such as cardenolides, terpenoids, phenolics, and alkaloids, which are listed as cytotoxic, antibacterial, antifungal, anthelmintic activities, as well as insect-repellent potential. Moreover, latex also contains diverse primary metabolites such as polypeptides and proteins [4]. Latex is produced by the plant immediately after mechanical damage or insect herbivory. According to Konno [4], specific plant tissues called laticifers are responsible for latex storage and typically look like a white milky sap. De Freitas et al. [5] found that *Thevetia peruviana* (Apocynaceae family) latex contains several plant defense proteins such as 120 kDa cysteine peptidase. Antibacterial activity against *Salmonella typhi*, *Klebsiella pneumoniae*, and *Streptococcus haemolyticus*, as well as antifungal activity toward filamentous fungus *Aspergillus niger* and the unicellular fungus *Candida albicans* were reported [6]. A recognized broad group of secondary metabolites in latex such as lignans, tannins, and coumarins, are characterized by antioxidant properties [7].

Ahmad et al. [8] mentioned the application of *T. peruviana* latex for treatment of decayed teeth, besides its use in chronic sores and ulcer therapy. Tinpun et al. [9] indicated that after cell injury, the application of *Jatropha curcas* latex extract encouraged wound healing with the appearance of antibacterial and antioxidant activity. Antiviral activity indicated that latex extracts had inhibitor potential not only for plant viruses but also toward human viruses including HPV, HSV-1, and HIV [10]. Recently, Gardin and Braga [11] used latex of *Chelidonium* majus to treat infection in humans exposed to severe acute respiratory syndrome coronavirus-2; the recovery level was observed after three days of latex administration. Gracz-Bernaciak et al. [12] assumed that latex proteins—alongside certain alkaloids—were responsible for antiviral activity. Furthermore, plant latexes were tested for cancer treatment against numerous cancer cells. Latex from *Euphorbia umbellata* exhibited cytotoxicity activity against HeLa and HRT-18 cells, melanoma cells (B16F10), and hepatocellular carcinoma. Additionally, in an in vivo study, minimization tumor mass in mice was observed with latex application [13]. Recently, numerous natural flavonoids such as epicatechin, luteolin, rutin, quercetin, naringenin, hesperidin, baicalin, and diosmin were applied for the management of SARS-CoV-2 [14]. In the current decade, molecular docking studies attracted the attention of biologists to assess the affinity of any natural or chemical molecules in relation to a particular biological target. Moreover, it permits to decrease the time and minimize the cost required to perform the same biological activity mechanisms [15,16]. The chemical composition of latex differs according to plant origin and development [17], therefore biological activities may differ not only according to the diversity of plants, but also to the plant origin as well as environmental conditions of cultivation. The present work aimed to study various biological activities of *T. peruviana* latex.

## 2. Materials and Methods

### 2.1. Collection of Thevetia peruviana Latex

*T. peruviana* latex was collected via cutting the stalk of fruits in tubes containing an appropriate volume of distilled and sterilized water (*v*/*v*) to the collection amount of latex. Voucher specimen of *T. peruviana* was positioned in the herbal collection, Science College, Al-Azhar University, Cairo, Egypt.

### 2.2. Flavonoid and Phenolic Analysis of Thevetia peruviana Latex by HPLC

Twenty µL of *T. peruviana* latex was mixed with 5 µL of 20% HCl as an aqueous solution, followed by heating for 90 min at 85 °C, then cooling and completing to 100 µL using ethanol. The mixture was filtrated through 0.45 µm membranes, then injected in HPLC-(Agilent 1100) (Agilent, Santa Clara, CA, USA), equipped with two LC pumps, a UV/Vis detector, and a C18 column (125 mm × 4.60 mm, 5 µm particle size) for flavonoid and phenolic content analysis. The obtained chromatograms were analyzed via the Agilent ChemStation. The mobile phase for phenolic acids separation was employed by two solvents including methanol and 1/25 of acetic acid/water (60:40 *v*/*v*), whereas equal volumes (50:50 *v*/*v*) of methanol/water (pH 2.8 were adjusted using phosphoric acid) were applied as mobile phase for flavonoids separation at isocratic flow level (one mL min^−1^) [18].

### 2.3. DPPH Radical Scavenging Protocol for Detecting the Antioxidant Activity

For detecting the antioxidant activity of latex, the free radical scavenging activity of latex was assayed via 1- diphenyl-2-picryl hydrazyl (DPPH). First, 1 mL solution of DPPH (0.1 mM) in ethanol was mixed with 3 mL of different doses of latex (3.9, 7.8, 15.62, 31.25, 62.5, 125, 250, 500, 1000 μg/mL) in ethanol prepared via dilution method. Then, the mixture was vigorously shaken, followed by standing for 30 min at 25 °C. The absorbance of the reaction mixture was measured at 517 nm using a spectrophotometer (UV-VIS milton roy). The inhibitory concentration 50% (IC_50_) of latex required to inhibit 50% of the DPPH free radical via log dose inhibition curve was calculated [19]. A number of dilutions ranging from 5 to 40 µg/mL of ascorbic acid as synthetic antioxidant (positive control) was prepared from a stock solution of 40 µg/mL. The following equation was used for calculating the percentage of the DPPH scavenging effect:$$\mathrm{DPPH}\;\mathrm{scavenging}\;\left(\%\right)=\frac{\;\mathrm{Control}\;\mathrm{reaction}\;\mathrm{absorbance}\;-\;\mathrm{Latex}\;\mathrm{reaction}\;\mathrm{absorbance}}{\mathrm{Control}\;\mathrm{reaction}\;\mathrm{absorbance}}\times 100$$

### 2.4. Antimicrobial Activity and Minimum Inhibitory Concentration (MIC) Detection

The inhibitor potential of the latex extract was performed against various bacteria, including *Bacillus subtilis*, *Staphylococcus aureus*, *Pseudomonas aeruginosa*, and *Escherichia coli*, against filamentous black fungus (*Mucor circinelloides*), and unicellular fungus (*Candida albicans*), by agar well diffusion test. Latex extract (100 uL) was poured in an agar well (6 mm) punctured with a sterile cork borer, then the growth agar media were inoculated by tested microorganisms, followed by refrigeration for 30 min, then incubation at 37 °C for 24 h (required for bacterial and yeast development) and 25 °C for 48 h (required for fungal growth). The diameter (mm) of the appeared inhibition zone around the well was recorded [20]. Positive control was applied using gentamycin and Ketoconazole as antibiotic and antifungal agents. Dimethyl sulfoxide (DMSO) for latex dissolution was also utilized as a control. MIC was detected by microdilution protocol via making serial dilutions of latex (dissolved in dimethylsulphoxide) in broth with use of a new pipette for each subsequent dilution step. The antimicrobial/broth solutions were dispensed into the plastic microdilution trays. The inoculum was prepared by making a saline suspension of bacterial/yeast colonies (2 × 10^8^ colony-forming units/mL) from 20 h old cultivated agar plate; 100 μL from the suspension was inoculated into each well. Then, the inoculated macrodilution tubes were incubated at 35 ± 2 °C for 18 h (required for bacteria development) or for 24 h (required for yeast growth). The MIC of latex was detected by the measure of OD at 600 nm compared with growth without treatment as a control [18].

### 2.5. Ultrastructure Examination of the Exposed Bacteria to Latex

Two treated species of bacteria by latex were examined by Transmission Electron Microscopy (TEM) (JEOL-JEM 1010) at 70 kV to show ultrastructure changes. Glutaraldehyde (3%) was used to fix the bacterial cells, then washed using phosphate buffer, followed by fixation at 25 °C in potassium permanganate solution. The prepared fixed samples were dehydrated for 15 min using gradual dilutions of ethanol ranging from 10% to 90%. Absolute ethanol was used for final dehydration of prepared samples for 30 min. The prepared specimens were infiltrated by a graded series through epoxy resin and acetone until they were in pure resin. Uranyl acetate was used to stain the ultrathin sections (stained twice) on copper grids, followed by lead citrate staining, and examination by TEM [18]. 

### 2.6. Coagulation Assay In Vitro

Classical coagulant assays including prothrombin time (PT) and activated partial thromboplastin time (APTT) tests were used for measuring the anticoagulant activity of latex. Human blood (nine parts) was drawn into one part of sodium citrate (3.2%), followed by centrifugation at 5000 rpm for 10 min and collection of supernatant. The blood was then used for PT and APTT assay. Latex at different dilutions was mixed with citrated normal human plasma followed by incubation at 37 °C for 3 min. The reagent of APTT was incubated for 3 min at 37 °C, then, 0.10 mL of the reagent was added to the reaction mixture of plasma and latex, then incubated at 37 °C for 5 min. At end of incubation time, pre-incubated 0.10 mL of CaCl_2_ (0.025 mol/L) at 37 °C for 3 min was added, then clotting time was recorded. As mentioned in the APPT assay, PT was also assayed, but a pre-incubated PT reagent (0.20 mL) at 37 °C for 3 min was added to the reaction mixture, and the clotting time was then recorded. The abovementioned technique was repeated, but only with the plasma (as control) and with heparin as positive control [21]. 

### 2.7. Preparation of Erythrocyte and Hypotonicity Induced Haemolysis

First, 3 mL of fresh whole blood was collected from healthy volunteers in heparinised tubes, and the blood samples were centrifuged (at 3000 rpm for 10 min). The collected pellets in the centrifuged tube were dissolved in an amount of normal saline solution identical to the volume of supernatant, then reconstituted (40% *v*/*v*) by suspension using isotonic buffer solution prepared from Na_3_PO_4_ buffer (10 mM) at pH 7.4. The reconstituted red blood cells (re-suspended supernatant) were used as such in isotonic solution (IS). 

Distilled water was used as a hypotonic solution (HS) for dissolving the utilized latex extract. Different levels of latex (100, 200, 400, 600, 800, and 1000 μg/mL) were added to 5 mL of HS (per dose) in centrifuge tubes. The prepared suspension of Erythrocyte (0.1 mL) was added to a tube containing HS with latex and mixed gently, then kept at 37 °C for 1 h, followed by centrifugation at 1300× *g* for 3 min. Absorbance (OD) at 540 nm was used to estimate the content of hemoglobin in the supernatant spectrophotometer (Milton Roy). Hemolysis % was calculated with regards to 100% hemolysis in distilled water [22]. The control tube contained 5 mL of distilled water, and another control contained 5 mL of 200 μg/mL of indomethacin as positive control. The provided formula was applied to determine the inhibition % of hemolysis (IH) by the latex:$$\mathrm{IH}\;\left(\%\right)=1-\frac{\;\mathrm{COD}2-\mathrm{OD}1}{\mathrm{OD}3-\mathrm{OD}1}\times 100$$
where OD1 and OD2 were the absorbance of latex in IS and in HS, respectively, and OD3 = absorbance of control in HS.

### 2.8. Cell Lines and Cytotoxicity Assay

Two cell lines of human breast cancer cells (MCF7) and prostate cancer cells (PC-3) were used to evaluate the antitumor activity of latex using a 3-(4,5-dimethylthiazol-2-yl)-2,5-diphenyltetrazolium bromide (MTT) approach. First, 5 mg of MTT was dissolved in 1 mL of phosphate buffer saline (PBS). The prepared cells (10^5^ cells/mL) were then inoculated in the 96-well tissue culture plate and incubated at 37 °C for one day to develop a whole monolayer sheet. Following the incubation time, the growth medium of cells was decanted from the plate wells, and the monolayer of cells was washed twofold with wash media. Different dilutions of latex were made in Gibco Human Plasma-like Medium (HPLM) supplemented with 2% serum. Each dilution (0.1 mL) was tested in different wells—leaving three wells as a control that received only maintenance medium—then incubated at 37 °C and scanned to notice any physical signs of toxicity. Next, 20 µL MTT solution was added to each well to mix the MTT into the media. The plate was placed on a shaking desk for 5 min around 150 rpm, then incubated at 37 °C under 5% CO_2_ for 1–5 h to permit the metabolization of MTT. MTT metabolic product was re-suspended in 200 µL of DMSO, followed by shaking to mix it into the solvent at 150 rpm for 5 min. The optical density (correlated directly with cell number) was measured at 560 nm and subtract background at 620 nm. Cell lines without plant extract in growth medium were studied as a control and treated with adriamycin as positive control [19]. The effect of latex was also evaluated on the normal cell lines, namely the Vero cell line (CCL-81). 

### 2.9. Molecular Docking

A molecular modeling study using the Molecular Operating Environment (MOE) module was conducted to explain the observed antibacterial and anticancer activity of the main detected compounds in latex extract, including chlorogenic acid and rutin. MOE’s BUILDER module was used to create the structural model, and the optimization conformational evaluations of the generated molecules were conducted in two steps. The geometry of the compounds was optimized using the semiempirical PM3 Hami-ltonian with Restricted Hartree-Fock (RHF) and RMS gradient of 0.05 Kcal/mol, as well as the integrated MOPAC 7.0 energy minimization tool [23].

The resulting model was then used in the MOE’s ‘Systematic Conformational Search’. To rank the compounds’ binding affinity to 7C7N, 3QUM, and 1JNX proteins, the binding free energy and bonds of hydrogen among the compounds and amino acid in (7C7N), (3QUM), and (1JNX) were utilized. Estimation of the bonds of hydrogen were performed by determining the length of hydrogen bond, moreover, RMSD (Root Mean Square Deviation) of the co-crystal ligand position compared to the docking pose was utilized in ranking. Both the RMSD as well as the mode of interaction of the native ligands within the crystal structure of *E. coli* DNA (7C7N), crystal structure of human prostate-specific antigen (PSA) in Fab sandwich with a high affinity and PCa-selective antibody (3QUM), and crystal structure of the BRCT repeat region from the breast cancer-associated protein (1JNX) receptor were utilized as a standard docked model.

### 2.10. Statistical Analysis

Experimental calculates were performed in replica, and therefore calculated as ± standard deviation (SD) and ± standard error (SE) means. GraphPad Prism^®^ (version 5.0) software was applied to obtain IC_50_ value of DPPH radical scavenging activity graphs 

## 3. Results and Discussion

### 3.1. Analysis of Phytoconstituents

The biological role of latex flavonoids and phenolic has been demonstrated in many studies, and the most recognized hypothesis comprises its contribution in different therapeutic potentials. Various flavonoids and phenolic compounds were detected in the collected latex of *T. peruviana* (Figure 1) by HPLC (Table 1 and Figure 2). Rutin and quercetin represent the most detected flavonoids with 11.45 and 7.15 µg/mL, respectively, followed by hisperdin 6.07 µg/mL. Additionally, 5.25 µg/mL of naringin was among the identified ingredient of latex. Different biological activities were recorded for flavonoids; among these activities, anti-inflammatory and antioxidant activities were associated to naringin [24]. Recently, Atefeh et al. [25] demonstrated the anticancer effect of rutin against different cancer cells including colon, lung, prostate, and breast. Moreover, lowering chemotherapy side effects and drug resistance were observed during rutin application with other anticancer drugs for tumor prevention. Srivastava et al. [26] suggest the potential role of quercetin for the treatment of breast cancer infection; specifically, its influence on healthy cells was very minimal or nonexistent. Scientists did not ignore the use of quercetin and other flavonoids against viruses. Colunga et al. [27] reported the antiviral activity of quercetin against several Coronaviridae family members, and used it in combination with vitamin C for the treatment of COVID-19 patients through the prevention of virus replication and assembly of proteins, as mentioned recently [28]. Researchers performed in vitro and in vivo experiments, and recognized rutin [25], kaempferol, and catechin [29] as nutritive additives applied to fight the proliferation of COVID-19. HPLC analysis reflected the presence of six phenolic compounds; among them, chlorogenic acid represented the main compound with a high concentration of 12.39 µg/mL, followed by syringenic acid 7.45 µg/mL (Table 1). Bio-functions of chlorogenic acid were reported, including blood clot degradation [30], and antioxidant, anti-inflammatory, hypoglycemic, and hypolipidemic benefits, besides other therapeutic properties [31]. It was also recently applied in vivo, and showed inhibitory action toward colorectal tumors in rats and humans [32]. In the present results, cinnamic acid and pyrogallol were detected with the lowest concentrations of 2.44 µg/mL and 2.78 µg/mL, respectively. Ferulic acid was detected in *Ficus carica* latex extracts and showed antiviral activities, stopping the propagation of European catfish virus (ECV-11), herpes simplex virus (HSV-1), and adenovirus [33]. It was also detected in *T. peruviana* latex with a concentration of 5.07 µg/mL (Table 1).

### 3.2. Antimicrobial Activity and Ultrastructure Changes

Antibiotics resistance has become a widespread phenomenon by many pathogenic microorganisms, and has reached unacceptable levels. This phenomenon highlights the urgent requirement for new types of antibiotics. The present outcome revealed stronger antibacterial action of latex against different bacteria species compared to standard antibiotics (Table 2 and Figure 3). *B. subtilis* followed by *E. coli* (inhibition zones of 31 and 30 mm, respectively) were the highest sensitive among the tested bacteria, while the lowest inhibition zone (23 mm) was noted for *P. aeruginosa*. Latex was more effective against *C. albicans* (IZ, 28 mm), and non-effective against black fungus (*Mucor circinelloides*). The obtained outcome was partially harmonized with Sibi et al. [34], where *T. peruviana* latex inhibited the growth of bacteria and fungi involving *P. aeruginosa, E. coli*, *S. pneumoniae*, *C. albicans*, and *Penicillium chrysogenum*. Proteolytic enzymes—besides other proteins—were identified in plant latex, as mentioned in previous studies, which lyse the component of the bacterial cell wall. The inhibitory action of the latex was supported by another study performed by Walsh et al. [35], as it increases the permeability of the cell membrane that leads to ion escape from the cell. The minimum value of MIC was recorded with *S. aureus* (7.8 µg/mL), followed by *B. subtilis* (10.2 µg/mL), while the maximum MIC (15.62 µg/mL) was recorded with *P. aeruginosa* (Table 2). Methanolic extract of *Thevetia neriifolia* latex was more efficient than other solvent extracts toward *K. pneumoniae*, *E. coli*, *Micrococcus luteus*, *Streptococcus pneumoniae*, *S. aureus*, *Bacillus cereus*, and *Lactobacillus acidophilus* growth with IZ 23.87, 22.13, 21.82, 27.45, 22.12, 28.36, and 30.74 mm, respectively. Moreover, there was a lower MIC in the case of *E. coli* and *S. aureus*, with concentrations 0.14 µg/mL and 0.24 µg/mL, respectively [36].

Ultrastructure studies by TEM reflected the presence of changes in *S. aureus* and *P. aeruginosa* treated with latex (Figure 4). *S. aureus* cells with exposure to latex resulted in significant alterations in bacterial cell shape at 7.8 µg/mL and at 15 µg/mL (Figure 4). The development of significantly damaged cells with unusual forms and an empty central zone previously filled by the nucleoid–dubbed “ghost cells”—was one of these modifications (arrows), in addition to the reduction of electron-dense materials, cytoplasm shrinkage, and the creation of vacuoles. On the other hand, the untreated (control) of *S. aureus* had no noticeable morphological or structural alterations according to TEM examination. Compared with control *P. aeruginosa* cells that had identical rod intact cells, latex-treated *P. aeruginosa* cells at 15.62 µg/mL had higher cytoplasmic shrinkage (or disappearance) with irregularities and ruptures of the cytoplasmic membrane (Figure 4), internal vacuolization, and complete deformation in shape were appeared in cells exposed to 30 µg/mL of latex. Scanning electron microscopy showed drastic changes in *Streptococcus agalactiae* and *Propionibacterium acnesbacterial* cell walls exposed to *Euphorbia antiquorum* latex [37].

### 3.3. Antioxidant Activity

Research for antioxidants additives, particularly of natural origin, has become a global demand in the medical and nutritional field. *T. neriifolia* latex showed good antioxidant activity depending on its applied concentration (Figure 5). DPPH scavenging activity at 500 and 1000 µg/mL was 69.5 and 77.3%, respectively, with IC_50_, 43.9 µg/mL. The lowest concentration of latex 1.95 showed the relative percentage of DPPH scavenging reached 26.4%. The antioxidant activity of latex was compared to the antioxidant activity of synthetic antioxidant as positive control with IC_50_, 13.87 ± 1.4 µg/mL. Similar results were also detected for other sources of latex such as *Ficus carica*, which displayed the promising antioxidant activity of 65.91% of DPPH scavenging [38]. Recently, Samrot and Sea [39] reported the antioxidant activity of *Artocarpus heterophyllus* latex, where DPPH scavenging activity was 55.17% at 500 μg/mL. Avoiding damages caused by reactive oxygen species, for example, hydrogen peroxide, superoxide anion, and hydroxyl radical, may represent one of the antioxidant mechanisms of latex, as mentioned previously [38,39].

### 3.4. Antihemolysis Activity 

Erythrocyte membrane stabilization represents one of the methods employed to discover the anti-inflammatory activity of any drug or substance. This method was used in the present study using a hypotonic conditions that induced lysis of cells. Hemolysis inhibition% was increased with increment latex concentrations, but reached 58.5% only at the highest concentration 1000 µg/mL (Figure 6). Hemolysis inhibition may be due to the presence of one or more natural ingredients present in the latex. The efficacy of latex was compared with indomethacin as positive control at 200 µg/mL, which showed the highest hemolysis inhibition reached 91.0%. The anti-inflammatory potential of *Calotropis procera* [40] and *T. peruviana* [41] latexes was reported previously. 

### 3.5. Anticoagulant Activity

A negligible increment in the PT and APPT value of blood samples was recorded after treatment with latex at all applied concentrations (Figure 7), where the recorded time was 11.9, 12, and 13.1 s of PT, and 29.2, 29.6, and 29.6 s of APPT at 25, 50, and 75 µg/mL, respectively. Unlike the effect of heparin (Figure 7), at a low concentration of 25 µg/mL, the PT and APPT were 94.6 and 117.7 s, and reached 138 and 155.8 s at a high concentration (75 µg/mL), respectively. The obtained results reflected the lack of anticoagulant properties of latex and may possess fibrinolytic activity that possibly stimulates the dissolution of clots. Results of Osoniyi and Onajobi [42] indicated that the procoagulant and anticoagulant activities were coordinated with *J. curcas* latex via PT and APPT assays. The current study exhibited findings that disagree with other studies [43,44], which reported the anticoagulant activities of *Carica papaya* and *J. gossypiifolia* latex. Khuraishi et al. [45] assessed the anticoagulant potential of latex collected from plants involving *T. peruviana*, *Euphorbia milii*, *E. pulcherimma*, *Artocarpus altilis*, and *J. malacophylla*, and recorded the highest anticoagulant activity in last two plants only. 

### 3.6. Anticancer Activity

Latex extract showed anticancer potential against PC-3 and MCF-7 (Table 3). The toxicity effect of the latex on the two cells was approximately similar, particularly at a concentration of 250 μg/mL or above, where the toxicity was 95.61, 96.93, and 97.11% against PC-3 and 96.71, 96.64, and 96.23% against MCF-7 at 250, 500, and 1000μg/mL, respectively. The obtained results reflected the two cell lines’ death in a concentration-dependent manner. The calculated IC_50_ was 40.31 μg/mL less in the case of MCF-7 than IC_50_ 48.26 μg/mL in case PC-3. Anticancer activity of other plant latexes was reported previously, such as latex of *Calotropis procera* [46] and *Himatanthus drasticus* [47]. *Passiflora alata* leaf latex exhibited cytotoxic effect against PC-3 cell lines with IC_50_ < 30 µg/mL [48]. The obtained antiproliferative effect of latex against PC-3 and MCF-7 was further documented and morphologically examined. It was obvious that cell morphology was altered in both types of cancer cells at all tested concentrations, compared with the untreated cells (Figure 8). The cells unexposed to latex appeared polygonal or spindle-shaped, while clear changes such as cell shrinkage, reduced cell density, and lack of cell adhesion were evident in cells exposed to 31.25 and 62.5 μg/mL of latex. At high concentrations of 125 and 250 μg/mL, the treated cells became more detached and changed to spherical or irregular forms. At 500 μg/mL, very few cells appeared to have drastic changes, indicating high detachment and entering into the death phase, perhaps due to disturbance of the nuclear membrane. Other reports found similar effects, but using *Calotropis procera* latex [49] and *Ficus religiosa* latex extract [50]. The results of anticancer activity of latex were compared with adriamycin as positive control, with IC_50_, 58.07 μg/mL against PC3, and 38.20 μg/mL against MCF-7. Moreover, the impact of latex on the Vero cell line (CCL-81) showed less pronounced effects with IC_50_, 123.25 μg/mL (data not tabulated) than on normal cells.

### 3.7. Molecular Docking Study

Furthermore, and as additional proof for the biological screening examinations, we performed a docking process for chlorogenic acid with *E.coli* DNA (7C7N), and rutin with human prostate-specific antigen (3QUM) and breast cancer-associated protein (1JNX), which were selected based on past research and literature [36,48], in order to look into the binding mode and conformation structure that performs a vital function in the interaction among proteins and compounds (Figure 9). 

From the docking, we predicted that the inhibitor (chlorogenic acid) has strong interaction with the active site bound to chain (A) of (7C7N) and its energy value (−6.0422 kcal mol^−1^) (Table 4). The prostate cancer protein PC-3 (PDB = 3QUM) interacted through amino acid pocket molecules with rutin via O47, O64, and O66 by donating its H atoms or accepting atoms of H (Table 4). Similarly, the breast protein MCF-7 (PDB = 1JNX) interacted with the rutin by donating diverse amino acid hydrogens, developing bonds of H with O11, O45, and O68 atoms (Table 4). The values of docking score for rutin against 3QUM and 1JNX receptors were (−6.8798 kcal mol^−1^) and (−6.9797 kcal mol^−1^), respectively. Increased interaction between compounds and receptors illustrate in vitro inhibitory actions. The best possible conformations of compounds inside the protein central activity are illustrated in Table 5; the quality and binding energy of the tested compounds with the molecules of the biological targets were documented via the consensus scoring function values. Molecular docking study was recently conducted using chlorogenic acid on *Proteus vulgaris* and human coronavirus (HCoV 229E) with a negative score of free binding energy appearing greater, demonstrating the utilization of chlorogenic acid as an antibacterial and antivirus agent [23].

## 4. Conclusions

The overall findings of the current research observed that *T. peruviana* latex is a prospective natural source of multiple pharmacological functions, demonstrating anti-inflammatory, antimicrobial, antioxidant, and anticoagulant activities. The findings will be valuable for future research in the fields of food supplements, food additives, and medication production. The molecular docking study’s energy ratings supports the biological activity of chlorogenic acid against *E. coli* and rutin against human prostate breast cancer cell lines.

## Figures and Tables

**Figure 1 molecules-27-03165-f001:**
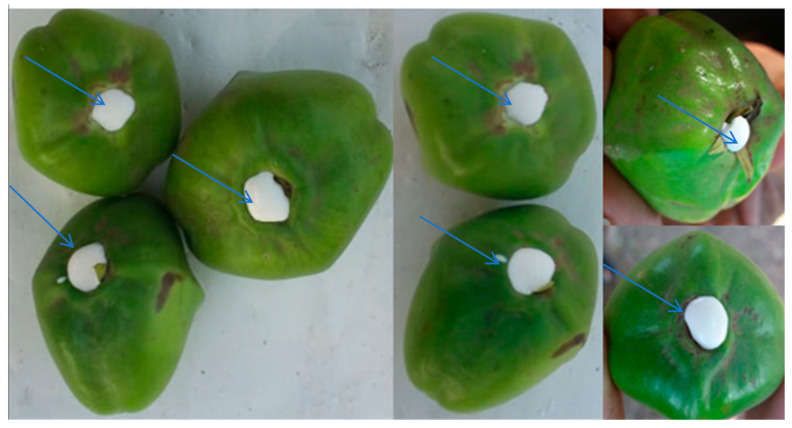
Collected latex (arrow direction to white latex secretion) from stipe of fruits.

**Figure 2 molecules-27-03165-f002:**
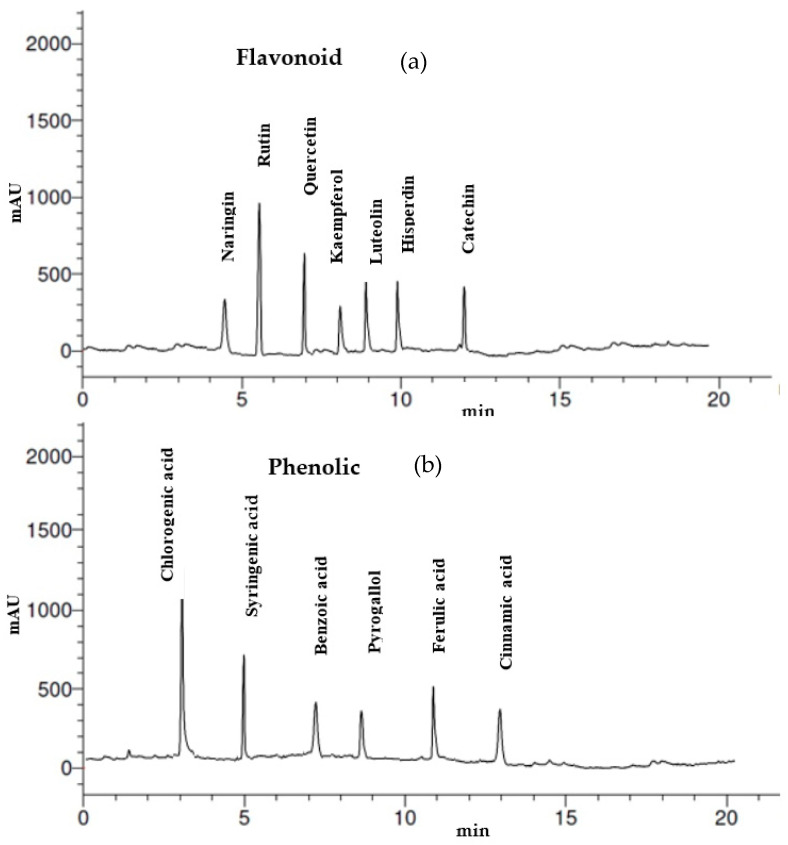
HPLC chromatograms of flavonoids (**a**) and phenolic (**b**) contents of latex.

**Figure 3 molecules-27-03165-f003:**
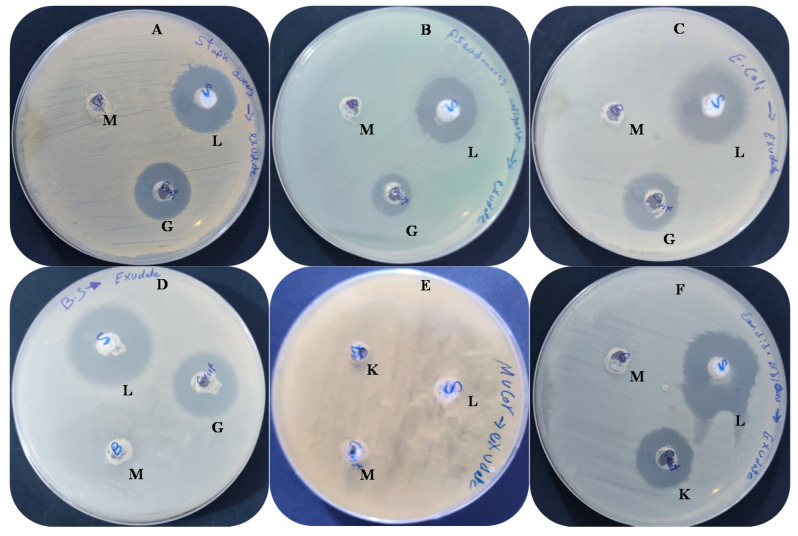
Antimicrobial activities against *S. aureus* (**A**), *P. aeruginosa* (**B**), *E. coli* (**C**), *B. subtilis* (**D**), *M. circinelloides* (**E**), and *C. albicans* (**F**); where antimicrobial activity represented by latex (L), methanolic (M), Gentamycin (G), Ketoconazole (K).

**Figure 4 molecules-27-03165-f004:**
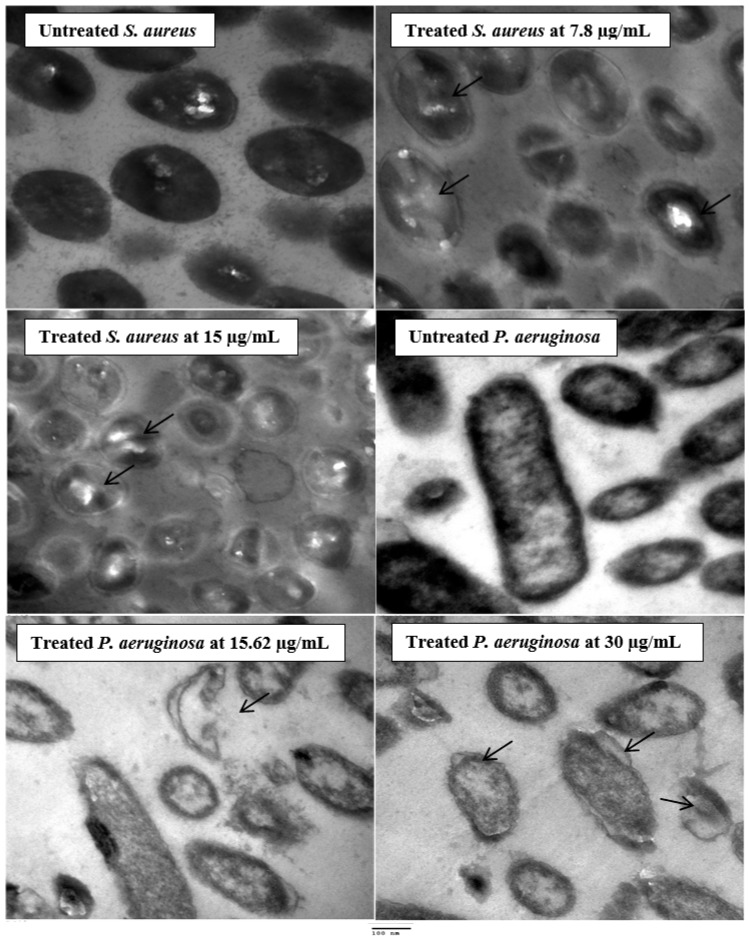
TEM micrographs of *S. aureus* and *P. aeruginosa* treated by latex. Scale Bar = 100 nm, 5000×.

**Figure 5 molecules-27-03165-f005:**
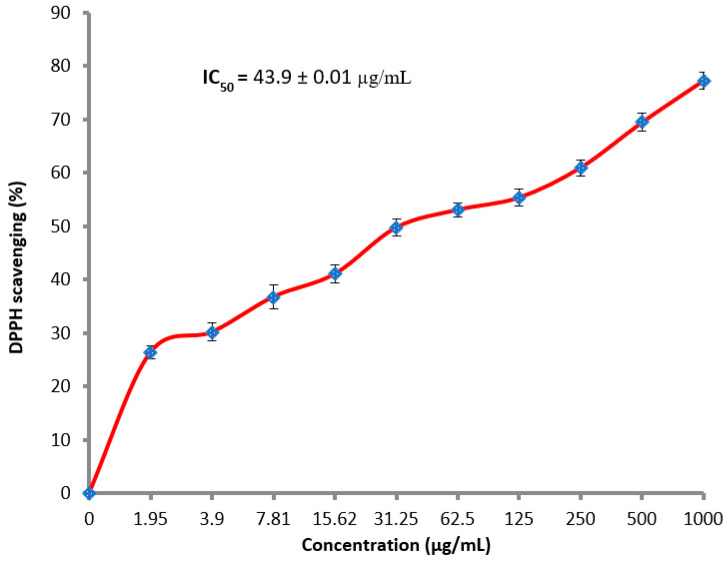
Antioxidant activity of latex.

**Figure 6 molecules-27-03165-f006:**
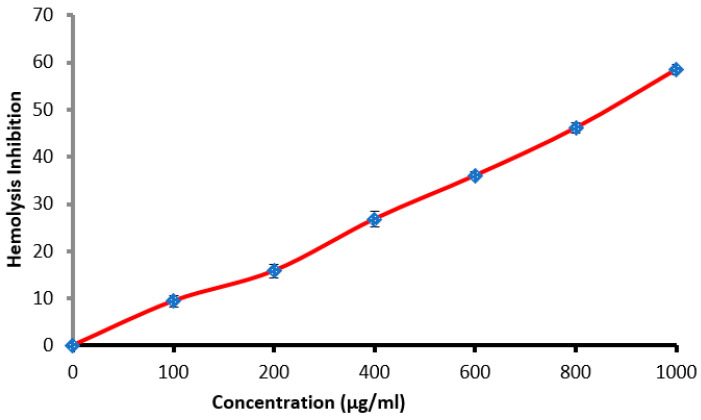
Hemolysis inhibition (%) at different concentration of latex.

**Figure 7 molecules-27-03165-f007:**
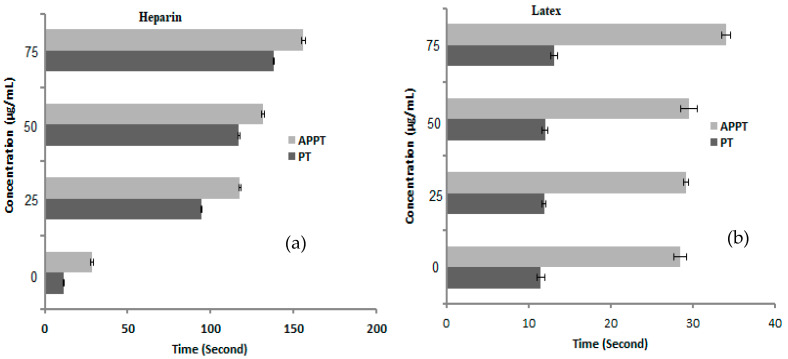
Anticoagulant activity of heparin (**a**) and latex (**b**) at different concentrations represented by Prothrombin time (PT) and Activated Partial thromboplastin time (APTT). Error bars ± standard deviation.

**Figure 8 molecules-27-03165-f008:**
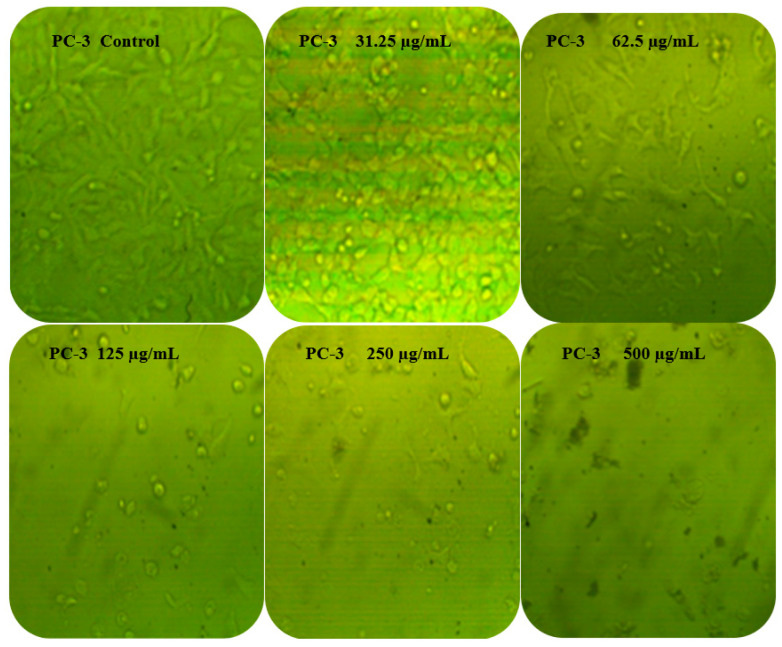
Effect of different concentrations of latex on morphological changes of PC-3 and MCF-7. Magnification, 40×.

**Figure 9 molecules-27-03165-f009:**
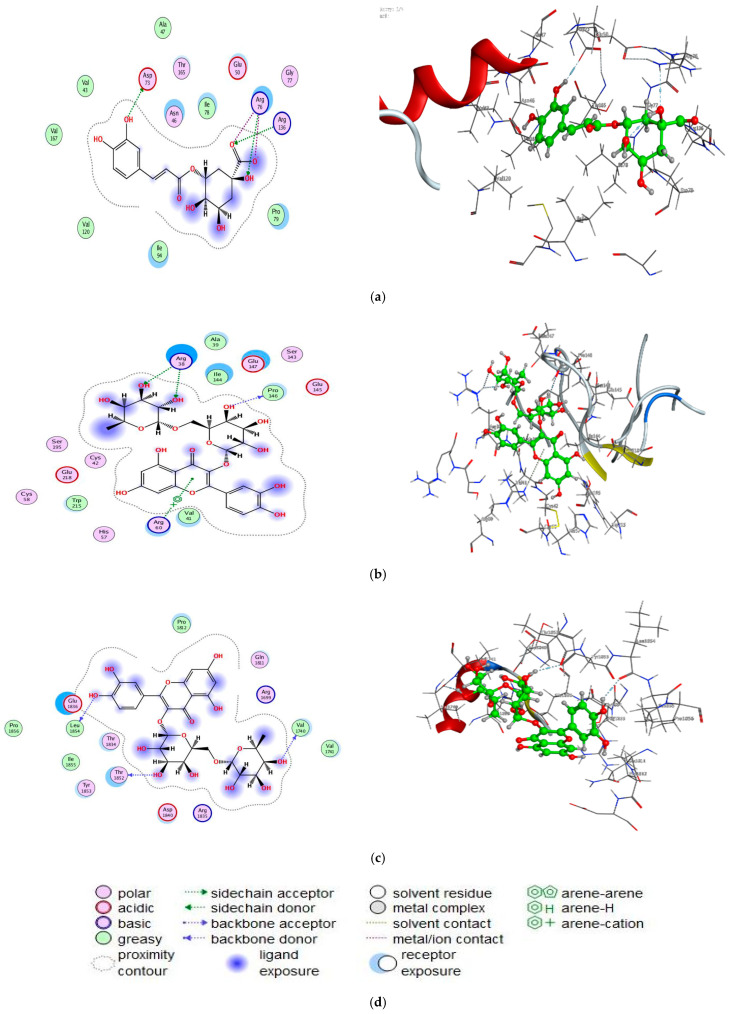
Docking interactions of certain compounds of latex (**a**) interaction between chlorogenic acid and active sites of 7C7N protein, (**b**) interaction between rutin and active sites of 3QUM protein, (**c**) interaction between rutin and active sites of 1JNX protein. (**d**) representative key for the types of interaction among rutin, chlorogenic acid, and proteins.

**Table 1 molecules-27-03165-t001:** Flavonoids and phenolic contents of *T. peruviana* latex.

Flavonoids			Phenolic Acids		
RT *	Compound	Concentration (µg/mL)	RT *	Compound	Concentration (µg/mL)
4.4	Naringin	5.25	3	Chlorogenic acid	12.39
5.3	Rutin	11.45	5	Syringenic acid	7.45
7.0	Quercetin	7.15	7	Benzoic acid	3.55
8.1	Kaempferol	4.65	9	Pyrogallol	2.78
9.0	Luteolin	5.02	11	Ferulic acid	5.07
10.0	Hisperdin	6.07	13	Cinnamic acid	2.44
12.0	Catechin	4.17			

* RT, retention time.

**Table 2 molecules-27-03165-t002:** Antimicrobial activity of *T. peruviana* latex extract against clinically important pathogens.

Tested Microorganisms		Inhibition Zone (mm)		MIC µg/mL
		Extract (100 µL)	Control *	
Gr + ve bacteria	*S. aureus*	25.33 ± 0.20	20.20 ± 0.23	7.80 ± 0.05
	*B. subtilis*	31.25 ± 0.33	25.10 ± 0.21	10.20 ± 0.12
Gr-ve bacteria	*E. coli*	30.33 ± 0.50	21.21 ± 0.20	12.51 ± 0.05
	*P. aeruginosa*	23.21 ± 0.40	15.02 ± 0.21	15.62 ± 0.04
Fungi	*C. albicans*	28.02 ± 0.05	23.04 ± 0.06	15.50 ± 0.33
	*M. circinelloides*	0.0 ± 0.00	0.0 ± 0.00	0.0 ± 0.00

* Ketoconazole against fungi, * Gentamycin against bacteria, Disc diameter: 6.0 mm loaded by 100 µL latex).

**Table 3 molecules-27-03165-t003:** Anticancer activity of latex against PC-3 and MCF-7.

Concentration μg/mL	PC-3				MCF-7			
	Mean O.D	SE	Viability%	Toxicity%	Mean O.D	SE	Viability%	Toxicity%
Control	0.554	0.010	100	0.00	0.476	0.010	100	0.00
31.25	0.39	0.008	69.80	30.20	0.30	0.006	62.11	37.89
62.5	0.15	0.011	26.71	73.29	0.13	0.008	26.75	73.25
125	0.06	0.009	11.43	88.57	0.02	0.001	3.43	96.57
250	0.02	0.004	4.40	95.61	0.02	0.001	3.29	96.71
500	0.017	0.001	3.07	96.93	0.02	0.001	3.36	96.64
1000	0.02	0.001	2.89	97.11	0.02	0.001	3.78	96.23
IC_50_	48.26 μg/mL				40.31 μg/mL			

**Table 4 molecules-27-03165-t004:** Interaction of active compounds with proteins.

Atoms/Ring	Chlorogenic Acid Interaction with 7C7N Protein			
	Receptor	Interaction	Distance	E (kcal/mol)
O 39	OD2 ASP 73 (A)	H-donor	2.79	−4.8
O 15	NH1 ARG 76 (A)	H-acceptor	3.00	−0.6
O 23	NH2 ARG 136 (A)	H-acceptor	3.13	−7.2
O 22	NH2 ARG 76 (A)	Ionic	3.05	−4.2
Atoms/Ring	Rutin interaction with 3QUM protein			
O 47	O PRO 146 (P)	H-donor	2.98	−0.6
O 64	NE ARG 38 (P)	H-acceptor	3.24	−1.2
O 66	NE ARG 38 (P)	H-acceptor	3.07	−0.5
6-ring	NH1 ARG 60 (P)	Pi-Cation	4.94	−0.5
Atoms/Ring	Rutin interaction with 1JNX protein			
O 11	O LEU 1854 (X)	H-donor	2.91	−1.6
O 45	O THR 1852 (X)	H-donor	2.95	−1.7
O 68	O VAL 1740 (X)	H-donor	2.95	−1.3

**Table 5 molecules-27-03165-t005:** The best possible conformations of compounds inside the protein central activity.

Compound	Receptor	Mol.	S	Rmsd_Refine	E_Conf	E_Place	E_Score 1	E_Refine	E_Score 2
Chlorogenic Acid	7C7N	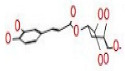	−6.0422	0.9533	−14.1250	−47.7348	−8.7382	−35.4672	−6.0422
Rutin	3QUM	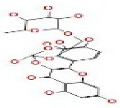	−6.8798	1.8680	140.3166	−92.8697	−11.6206	−42.6250	−6.8798
Rutin	1JNX	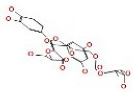	−6.9797	1.5443	128.9704	−27.3114	−8.6365	−43.6968	−6.9797

Where, S = final score, which is the score of the last stage that was not set to none; Rmsd = the root mean square deviation of the pose, in Å, from the original ligand; Rmsd_Refine = the root mean square deviation between the pose before refinement and the pose after refinement; E_Conf = the energy of the conformer; E_Place = score from the placement stage; E_Score 1, E_Score 2 = score from rescoring stages 1 and 2, respectively; E_Refine = score from the refinement stage, calculated to be the sum of the van der Waals electrostatics and solvation energies, under the Generalized Born solvation model (GB/VI).

## Data Availability

All data that support the findings of this study are available within the article.

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
