# Peer review of "Anticancer, Anticoagulant, Antioxidant and Antimicrobial Activities of Thevetia peruviana Latex with Molecular Docking of Antimicrobial and Anticancer Activities"

_molecules, 2022, doi:10.3390/molecules27103165_

Round 1
Reviewer 1 Report
After reviewing the mansucript entitled: Anticancer, anticoagulant and Antimicrobial Activities of Thevetia peruviana Latex with Molecular Docking of antimicrobial and anticancer activities" and corrected according to my comments in the first roud, this articles could be accepted in its current form.
Reviewer 2 Report
The article entitled “Anticancer, Anticoagulant and Antimicrobial Activities of Thevetia peruviana Latex with Molecular Docking of Antimicrobial and Anticancer activities” describes the process of isolation and characterization of latex secreted by Thevetia peruviana. Moreover, the authors have conducted a set of studies to evaluate the anticancer, anticoagulant, and antimicrobial activities of such compounds, including in silico molecular docking.
Positive points:
The authors have addressed an interesting topic and characterized a natural subproduct in terms of its molecular properties and possible health, agriculture, and industrial applications. Moreover, the authors have conducted several techniques for characterization and testing.
Negative points:
However, I have some concerns regarding the in-silico approach used in this project.
- The authors must provide a better justification for the selection of just these 3 structures to perform the molecular docking. According to the properties identified for the molecules studied, there are many interesting targets to investigate.
- Topic 3.6.: What is the impact of the tested compounds on (nontumoral) normal cells? At which concentration does it become toxic?
- It is not clear if the authors conducted any kind of validation of the molecular docking protocol employed. For example, trying to re-dock the co-crystallographic ligands and check the resulting pose. This is essential to ensure that the docking protocol is describing adequately the interaction within the binding site.
- Molecular dynamics simulations followed by free energy calculations would considerably increase the meaningfulness of the results obtained for the interaction of these compounds with the respective proteins.
- How do these interaction results compare with other know binding molecules, including other known inhibitors?
Other minor issues:
- The abstract is too long. It should be more concise and clear.
- There should be a space between values and units, e.g., 11.45µg/mL, 11.9s…
- The organism's scientific names should be placed in italic, e.g., Candida albicans, E. coli…
- There are a lot of formatting issues regarding the text of the topic 2.9.
- Reference 23 seems not to be related to the MOPAC 7.0 energy minimization tool.
- Line 214: the PDB code 7C7N is not the structure of E.coli DNA
- Table 5: the authors need to include a description for each parameter presented. It is not clear what S, E_conf, E_place, E_score*… mean.
Reviewer 3 Report
The manuscript by Al-Rajhi et al describes the evaluation of diverse biological activities of Thevetia peruviana latex. After carefully analysing the manuscript, I could not recommend its publication in Molecules journal.
The research should be conducted and the manuscript should be rewritten in a more logical way.
There are many important aspects to be considered, some of which are listed below :
- The whole manuscript needs extensive English revisions, some terms are not adequate used and a lot of statements are misleading.
- The title is not in accordance with all activities described in the manuscript (for example, the antioxidant activity).
- The methods of some determinations are missing from the Material and Methods section (such as, the anticancer activity).
Round 2
Reviewer 2 Report
The authors have addressed my main concerns about the manuscript.
Reviewer 3 Report
The current version of the manuscript can be accept in present form.
This manuscript is a resubmission of an earlier submission. The following is a list of the peer review reports and author responses from that submission.
Round 1
Reviewer 1 Report
Thank you for allowing me to review this manuscript entitled: “Molecular Docking Studies for Anticancer and Antimicrobial Activities of Latex with Therapeutic Evaluating In Vitro”. Authors did a good, and planned paper mixing docking analysis on a specific target protein in breast cancer and antimicrobial activity in vitro, the discussion is supported by the HPLC analysis to quantify the metabolites and to relate with the biological activity.
However, there are minor comments:
In title I suggest including the anticoagulant activity. Authors included as part of other test which increase the validation in the future. Furthermore include the name of the plant (Scientific name).
Authors should include the voucher identification of the plant by an herbarium in the methodology.
In table 2. I cannot to appreciate the statistical analysis and the repetitions plus standard deviation. Did you analyze it?
In Figure 3, I suggest improving the legends for instance include the name in each plate, many letter is hard to understand its interpretation as well as the used concentrations in each well.
In figure 7, and 8 the statistical analysis should be mentioned.
Although you analyze the phytochemical components by HPLC for phenolic compounds. Why did not you analyze the presence of cardenolides? They could be toxic in high content. I suggest including this limitation in your discussion and to mention the toxicity of T. peruviana.
Reviewer 2 Report
The manuscript by Al-Rajhi et al. presents a summation of various experimental data regarding the screening of different bioactivities of the latex of Thevetia peruviana fruits. Although some findings are interesting, the manuscript in its current form lacks in proper design of the study, clear presentation of the results and the scientific rationale behind certain experimental steps.
- The title of the manuscript should be changed. Molecular docking is only a small part of the research work and I find hard to see the connectivity with the rest of the results that are being presented to stress the emphasis on molecular docking studies.
- Abstract and Introduction require better organization of structure.
- Table 2 – Present the data as AVERAGE ± S.D Statistical analysis is missing.
- Figure 3 should be provided as supplementary data
- Figure 4. TEM micrographs are not of high quality. Provide sharper images and place them in order of three images based on the conditions of treatment (1. Control – untreated, 2. Treated with low concentration, 3. Treated with higher concentration) in two panels (panel A – S. aureus, panel B – P. aeruginosa)
- The authors used only DPPH assay for evaluating the antioxidant activity of latex. No positive control was used. Error bars are missing in Figure 5. Additional antioxidant assays shou be performed to better describe the antioxidant activity. Discuss on the potential mechanism(s) of the antioxidant action of latex?
- The authors investigated the antiproliferative activity of the latex extract in PC3 and MCF7 cells. Positive control is also missed in these studies. The antiproliferative activity of the latex extract should be tested in normal phenotype prostate and breast cells before a potential anticancer potential is claimed. Data in Table 3 should be presented as AVERAGE ± S.D. More concentrations should be tested between 0 – 250 mg/ml (at least 8-10 total different concentrations) for a more accurate estimation of the EC50. Please use the term Efficient Concentration (EC50) rather than Inhibitory Concentration (IC50) for the description of the antiproliferative activity. EC50 values should be presented as EC50±S.D.
- Combine figures describing anticoagulant activity 7 & 8 in one. Statistical analysis is also missing in these data.
- Figure 9. Images are not of high analysis. Provide the scale of the magnification.
- Section 3.7. To my opinion the necessity and the scientific rationale behind the molecular docking studies are not appropriately explained. Why only chlorogenic acid and rutin were selected for such studies? The latex extract is a mixture of various compounds with very diverse biological activities. Further experimental validation of the conclusions of the molecular docking studies is required.
- Conclusions need to be restructured. In the conclusion section it is mentioned that T. peruviana latex is a prospective natural source of anti-inflammatory activity (???) No such data exist in the manuscript to support such a conclusion. The conclusions on the molecular docking studies that are in good agreement with the experimental data are not supported.
Reviewer 3 Report
The first major concern is the plant’s name.
Throughout the paper I cannot find the complete name of the plant used ‪Thevetia peruviana. The plant was not identified, no voucher was deposited
As far as I found in the literature, the commonly used name for this plant is Cascabela thevetia (L.) Lippold, this is why a botanist must investigate the plant’s origin and correct nomination.
According to Plants of The world, this plant is not native or introduced to the two countries the authors are affiliated from (Egypte and ‪Saudi Arabia).
This plant is poisonous, so it is not a plant that people will spread and cultivate.
Many doubts raised about the plant material.
Title
The title is not appropriate, I don’t know if all authors reviewed this before submission.
As far as I saw in the paper, the molecular docking study is about the antimicrobial, the in vitro study is about the anticancer, antimicrobial and other not cited activities such as antioxidant and hemolytic activity
Latex of what?
Abstract
‪I never saw an abstract starting with results !‪
Even if it is an unstructured abstract format, it should contain the following points one after the other. Background, aims, methods, results, and conclusion
The background is missing.
The aims are not presented.
Methods are missing.
Results are all that you have written in the abstract
The conclusion is missing
Introduction
I’m guessing this part is a word to word translation from another language. I have no other explanation for the level of English in this part.
You did not mention the toxicity of the used plant nor the novelty of your work.
It is just a pack of sentences without any clear point.
Material and methods
The plant material description lacks all important info that confirms the plant's identity.
The Anticancer methods are entirely missing!
HPLC protocol is incomplete
The whole section is questionable!
Again the language of this part is a shame.
I believe that the common thing to do in research and collaborations is work along with all authors in order to conduct the study and write the paper properly.
As far as I saw, even the full name of the authors was not reviewed before the submission! missing last names!
Please reconsider the way you work and be respectful to the research field.
I cannot point out to all basic points that should already been fixed before the submission in order to let me try a critical reading of your work
‪
The flaws and English of the paper are not something you can fix with a revision. Please refer to a senior researcher to correct the writing, the methodology, the technical and scientific terms.